

# 2D facial landmark localization method for multi-view face synthesis image using a two-pathway generative adversarial network approach

Mahmood H.B. Alhlffee[1], Yea-Shuan Huang[2] and Yi-An Chen[2]

[1] College of Computer Science and Electrical Engineering, Chung-Hua University, Hsinchu, Taiwan
[2] Department of Computer Science and Information Engineering, Chung-Hua University, Hsinchu, Taiwan

Corresponding authors
Mahmood H.B. Alhlffee,
mahmood.bidir1985@gmail.com
Yea-Shuan Huang,
yeashuan@chu.edu.tw

## ABSTRACT

One of the key challenges in facial recognition is multi-view face synthesis from a single face image. The existing generative adversarial network (GAN) deep learning methods have been proven to be effective in performing facial recognition with a set of pre-processing, post-processing and feature representation techniques to bring a frontal view into the same position in-order to achieve high accuracy face identification. However, these methods still perform relatively weak in generating high quality frontal-face image samples under extreme face pose scenarios. The novel framework architecture of the two-pathway generative adversarial network (TP-GAN), has made commendable progress in the face synthesis model, making it possible to perceive global structure and local details in an unsupervised manner. More importantly, the TP-GAN solves the problems of photorealistic frontal view synthesis by relying on texture details of the landmark detection and synthesis functions, which limits its ability to achieve the desired performance in generating high-quality frontal face image samples under extreme pose. We propose, in this paper, a landmark feature-based method (LFM) for robust pose-invariant facial recognition, which aims to improve image resolution quality of the generated frontal faces under a variety of facial poses. We therefore augment the existing TP-GAN generative global pathway with a well-constructed 2D face landmark localization to cooperate with the local pathway structure in a landmark sharing manner to incorporate empirical face pose into the learning process, and improve the encoder-decoder global pathway structure for better representation of facial image features by establishing robust feature extractors that select meaningful features that ease the operational workflow toward achieving a balanced learning strategy, thus significantly improving the photorealistic face image resolution. We verify the effectiveness of our proposed method on both Multi-PIE and FEI datasets. The quantitative and qualitative experimental results show that our proposed method not only generates high quality perceptual images under extreme poses but also significantly improves upon the TP-GAN results.

## INTRODUCTION

Face recognition is one of the most commonly used biometric systems for identifying individuals and objects on digital media platforms. Due to changes in posture, illumination, and occlusion, face recognition faces multiple challenges. The challenge of posture changes comes into play when the entire face cannot be seen in an image. Normally, this situation may happen when a person is not facing the camera during surveillance and photo tagging. In order to overcome these difficulties, several promising face recognition algorithms based on deep learning have been developed, including generative adversarial networks (GANs). These methods have been shown to work more efficiently and accurately than humans at detection and recognition tasks. In such methods, pre-processing, post-processing, and multitask learning or feature representation techniques are combined to provide high accuracy results on a wide range of benchmark data sets (*Junho et al., 2015*; *Chao et al., 2015*; *Xi et al., 2017*; *Jian et al., 2018*). The main hurdle to these methods is multi-view face synthesis from a single face image (*Bassel, Ilya & Yuri, 2021*; *Chenxu et al., 2021*; *Yi et al., 2021*; *Hang et al., 2020*; *Luan, Xi & Xiaoming, 2018*; *Rui et al., 2017*). Furthermore, a recent study (*Soumyadip et al., 2016*) emphasized that compared with frontal face images with yaw variation less than 10 degrees, the accuracy of recognizing face images with yaw variation more than 60 degrees is reduced by 10%. The results indicate that pose variation continues to be a challenge for many real-world facial recognition applications. The existing approaches to these challenges can be divided into two main groups. In a first approach, frontalization of the input image is used to synthesize frontal-view faces (*Meina et al., 2014*; *Tal et al., 2015*; *Christos et al., 2015*; *Junho et al., 2015*), meaning that traditional facial recognition methods are applicable. Meanwhile, the second approach focuses on learning discriminative representations directly from non-frontal faces through either a one-joint model or multiple pose-specific models (*Omkar, Andrea & Andrew, 2015*; *Florian, Dmitry & James, 2015*). It is necessary to explore the above approaches in more detail before proceeding. For the first approach, the conventional approaches often make use of robust local descriptors, (such as *John, 1985*; *Lowe, 1999*; *Ahonen, Hadid & Pietikäinen, 2006*; *Dalal & Triggs, 2005*), to account for local distortions and then adapt the metric learning method to achieve pose invariance. Moreover, local descriptors are often used (*Kilian & Lawrence, 2009*; *Tsai-Wen et al., 2013*) approaches to eliminate distortions locally, followed by a metric learning method to prove pose invariance. However, due to the tradeoff between invariance and discriminability, this type of approach is relatively weak in handling images with extreme poses. A second approach, often known as face rotation, uses one-joint models or multiple pose-specific models to learn discriminative representations directly from non-frontal faces. These methods have shown good results for near-frontal face images, but they typically perform poorly for profile face images because of severe texture loss and artifacts. Due to this poor performance, researchers have been working to find more effective methods to reconstruct positive facial images from data (*Yaniv et al., 2014*; *Amin & Xiaoming, 2017*; *Xi et al., 2017*). For instance, *Junho et al. (2015)*, adopted a multi-task model to improves identity preservation over a single task model from paired training data. Later on (*Luan, Xi & Xiaoming, 2017*; *Rui et al., 2017*),

their main contribution was a novel two-pathway GAN architecture tasks for photorealistic and identity preserving frontal view synthesis starting from a single face image. Recent work by *Bassel, Ilya & Yuri (2021)*, *Chenxu et al. (2021)* and *Yi et al. (2021)* has demonstrated advances in the field of face recognition. During pose face transformation, however, some of the synthetic faces appeared incomplete and lacked fine detail. So far, the TP-GAN (*Rui et al., 2017*) has made significant progress in the face synthesis model, which can perceive global structure and local details simultaneously in an unsupervised manner. More importantly, TP-GAN solves the photorealistic frontal view synthesis problems by collecting more details on local features for a global encoder–decoder network along with synthesis functions to learn multi-view face synthesis from a single face image. However, we argue that TP-GAN has two major limitations. First, it is critically dependent on texture details of the landmark detection. To be more specific, this method focuses on the inference of the global structure and the transformation of the local texture details, as their corresponding feature maps, to produce the final synthesis. The image visual quality results indicate that these techniques alone have the following deficiencies: A color bias can be observed between the synthetic frontal face obtained by TP-GAN method and the input corresponding to non-frontal input. In some cases, the synthetic faces are even incomplete and fall short in terms of fine detail. Therefore, the quality of the synthesized images still cannot meet the requirements for performing specific facial analysis tasks, such as facial recognition and face verification. Second, it uses a global structure, four local network architectures and synthesis functions for face frontalization, where training and inference are unstable under large data distribution, which makes it ineffective for synthesising arbitrary poses. The goal of this paper is to address these challenges through a landmark feature-based method (LFM) for robust pose-invariant facial recognition to improve image resolution under extreme facial poses.

In this paper, we make the following contributions:

The LFM is a newly introduced method for the existing generative global pathway structure that utilizes a 2D face landmark localization to cooperate with the local pathway structure in a landmark sharing manner to incorporate empirical face pose into the learning process. LFM of target facial details provides guidance to arbitrary pose synthesis, whereas the four-local patch network architecture remains unchanged to capture the input facial local perception information. The LFM provides an easy way for transforming and fitting two-dimensional face models in order to achieve target pose variation and learn face synthesis information from generated images.

In order to better represent facial image features, we use a denoising autoencoder (DAE) to modify the structure of the generator's global-path encoder and decoder. The goal of this modification is to train the encoder decoder with multiple noise levels so that it can learn about the missing texture face details. Adding noise to the image pixels causes them to diffuse away from the manifold. As we apply DAE to the diffused image pixels, it attempts to pull the data points back onto the manifold. This implies that DAE implicitly learns the statistical structure of the data by learning a vector field from locations with no data points back to the data manifold. As a result, encoder–decoders must infer missing pieces and retrieve the denoised version in order to achieve balanced learning behavior.

We optimize the training process using an accurate parameter configuration for a complex distribution of facial image data. By re-configuring the parameters (such as the learning rate, batch size, number of epochs, *etc.*), the GAN performance can be better optimized during the training process. Occasionally, unstable "un-optimized" training for the synthetic image problem results in unreliable images for extreme facial positions.

## RELATED WORK

In this section, we focus on the most recent studies which are related to the multi-view face synthesis problem using deep learning approaches. The deep learning approaches including face normalization, generative adversarial network and facial landmark detection, are reviewed.

### Face normalization

Face normalization, or multi-view face synthesis from a single face image, is a unique challenge for computer vision systems due to its ill-posed problem. The existing solutions to address this challenge can be classified into three categories: 2D/3D local texture warping methods (*Tal et al., 2015*; *Xiangyu et al., 2015*), statistical methods (*Christos et al., 2015*; *Li et al., 2014*), and deep learning methods (*Xi et al., 2017*; *Luan, Xi & Xiaoming, 2017*). *Tal et al. (2015)*, employed a single 3D reference surface for all query faces in order to produce face frontalization. *Xiangyu et al. (2015)* employed a pose and expression normalization method to recover the canonical-view. *Christos et al. (2015)*, proposed a joint frontal view synthesis and landmark localization method. *Li et al. (2014)*, concentrated on local binary pattern-like feature extraction. *Xi et al. (2017)* proposed a novel deep 3DMM-conditioned face frontalization GAN in order to achieve identity-preserving frontalization and high-quality images by using a single input image with a 90° face pose. *Luan, Xi & Xiaoming (2017)* proposed a single-pathway framework called the disentangled representation learning-generative adversarial network (DR-GAN) to learn identity features that are invariant to viewpoints, *etc.*

### Generative adversarial networks (GANs)

The GAN is one of the most interesting research frameworks that is used for deep generative models proposed by *Ian et al. (2014)*. The theory behind the GAN framework can be seen as a two-player non-cooperative game to improve the learning model. A GAN model has two main components, generator (G) and discriminator (D). G generates a set of images that is as plausible as possible in order to confuse the D, while the D works to distinguish the real generated images from the fake. The convergence is achieved by alternately training them. The main difference between GANs and traditional generative models is that GANs generate whole images rather than pixel by pixel. In a GAN framework, the generator consists of two dense layers and a dropout layer. A normal distribution is used to sample the noise vectors and feed them into the generator networks. The discriminator can be any supervised learning model. GANs have been proven effective for a wide range of applications, such as image synthesis (*Rui et al., 2017*; *Yu et al., 2018*; *Yu et al., 2020*), image super-resolution (*Christian et al., 2017*), image-to-image translation (*Jun-Yan et al., 2017*),

*etc.* Several effective GAN models have been proposed to cope with the most complex unconstrained face image situations, such as changes in pose, lighting and expression. For instance, *Alec, Luke & Soumith (2016)* proposed a deep convolutional GAN to integrate a convolutional network into the GAN model to achieve more realistic face image generation. *Mehdi & Simon (2014)* proposed a conditional version of the generative adversarial net framework in both generator and discriminator. *Augustus, Christopher & Jonathon (2017)* presented an improved version of the Cycle-GAN model called "pixel2pixel" to handle the image-to-image translation problems by using labels to train the generator and discriminator. *David, Thomas & Luke (2017)* proposed a boundary equilibrium generative adversarial network (BE-GAN) method, which focuses on the image generation task to produce high-quality image resolution, *etc.*

### Facial landmark detection

The face landmark detection algorithm is one of most successful and fundamental components in a variety of face applications, such as object detection and facial recognition. The methods used for facial landmark detection can be divided into three major groups; holistic methods, constrained local methods, and regression-based methods. In the past decade, deep learning models have proven to be a highly effective way to improve landmark detection. Several existing methods are considered to be good baseline solutions to the 2D face alignment problem for faces with controlled pose variation (*Xuehan & Fernando De la, 2013*; *Georgios, 2015*; *Xiangyu et al., 2015*; *Adrian & Georgios, 2017*). *Xuehan & Fernando De la (2013)*, proposed supervised descent method, which learns the general descent directions in a supervised manner. *Georgios (2015)*, in their method, a sequence of Jacobian matrices and hessian matrices is determined by using regression. *Xiangyu et al., 2015* proposed a 3D model with cascaded convolutional neural network to solve the self-occlusion problem. *Adrian & Georgios (2017)*, proposed a guided-by-2D landmarks convolutional neural network that converts 2D annotations into 3D annotations, *etc.*

We can summarize some important points from our related work. Despite the fact that the existed methods produced good results on the specific face image datasets for which they were designed and provided robust alignment across poses, they are difficult to replicate if they are applied alone to different datasets. This is especially true for tasks like facial normalization or other face synthesis tasks, where deep structure learning methods still fail to generate high-quality image samples under extreme pose scenarios, which results in significantly inferior final results.

## PROPOSED METHOD

In this section, we shall first briefly describe the existing TP-GAN architecture and then describe our proposed LFM method in detail.

### TP-GAN architecture

Based on the structure shown in Fig. 1, the TP-GAN framework architecture consists of two stages. The first stage is a generator of two-pathways CNN $G_{\theta_G}$ that is parameterized by $\theta_G$. Each pathway has encoder–decoder $\{G_{\theta_E}, G_{\theta_D}\}$ structure and combination of loss functions,

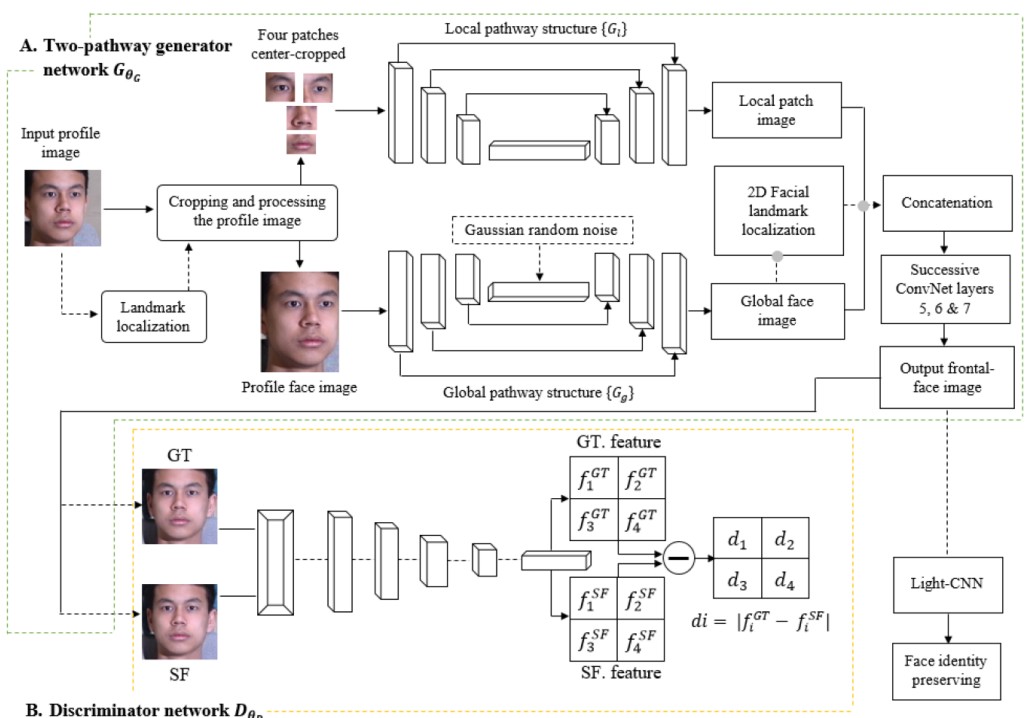

**Figure 1 The structure of TP-GAN.** The final output was obtained by integrating the global pathway with a 2D facial landmark localization to collaborate with the local pathway in a landmark sharing fashion. The dataset was downloaded from the official TP-GAN GitHub page: https://github.com/HRLTY/TP-GAN.

a local pathway $\{G_{\theta_E^l}, G_{\theta_D^l}\}$ of four landmark patch networks $G_{\theta_i^l}, i \in \{0, 1, 2, 3\}$ to capture the local texture around four facial landmarks, and one global network $\{G_{\theta_E^g}, G_{\theta_D^g}\}$ to process the global face structure. Furthermore, the bottleneck layer ($G_g$), which is the output of $G_{\theta_E^g}$, is typically used for classification tasks with the cross-entropy loss $L_{\text{cross-entropy}}$. A global pathway helps to integrate facial features with their long-range dependencies and, therefore, to create faces that preserve identities, especially in cases of faces with large pose angles. In this way, we can learn a richer feature representation and generate inferences that incorporate both contextual dependencies and local consistency. The loss functions, including pixel-wise loss, symmetry loss, adversarial loss, and identity preserving loss, are used to guide an identity preserving inference of frontal view synthesis. The discriminator $D_{\theta_D}$ is used to distinguish real facial images $I^F$ or 'ground-truth (GT) frontal view' from synthesized frontal face images $G_{\theta_G}(I^P)$ or 'synthesized-frontal (SF) view'. A second stage involves a light-CNN model that is used to compute face dataset's identity-preserving properties. For a more detailed description, see *Rui et al. (2017)*.

## LFM for generative global pathway structure

To the best of our knowledge, this is the first study to integrate an LFM with the existing TP-GAN global pathway structure for training and evaluation purposes. In this work, we exploit a landmark detection mechanism (*Adrian & Georgios, 2017*) that proposed for 2D-to-3D facial landmark localization to help our model obtain a high quality frontal-face

image resolution. Face landmarks are the most compressed representation of a face that maintains information such as pose and facial structure. There are many situations where landmarks can provide advanced face-related analyses without using whole face images. The landmark method used in this study was explored at (*Xi et al., 2017*; *Jian et al., 2018*; *Xing, Sindagi & Vishal, 2018*). These methods can achieve high accuracy of face alignment by cascaded regression methods. Methods like these work well when particular poses are chosen without taking other factors into consideration, such as facial characteristics. We found that facial characteristics can play an imperative role in improving the results of the current state of the art. By adding landmarks to augment the synthesized faces, recognition accuracy will be improved since these landmarks rely on generative models to enhance the information contained within them. The process for generating facial images is shown in Fig. 1. We will discuss our Fig. 2 architecture in the subsequent paragraph. We perform a face detection to locate the face in the Multi-PIE and FEI datasets. The face detection can be achieved by using a Multi-Task Cascade CNN through the MTCNN library (*Kaipeng et al., 2016*). After that, cropping and processing of the profile image. A local pathway of four landmark patch networks $G_{\theta_i^l}, \{i \in 0, 1, 2, 3\}$ to capture the local texture around four facial landmarks. Each patch learns a set of filters for rotating the center-cropped patch (after rotation, the facial landmarks remain in the center) to its corresponding frontal view. Then, we used a multiple feature map to combine the four facial tensors into one. Each tensor feature is placed at a "template landmark location" and a max-out fusing strategy is used to ensure that stitches on overlapping areas are minimized. Then, a 2D zero padding technique is used to fill out the rows and columns around the template landmark location with zeros. Nevertheless, local landmark detection alone cannot provide accurate texture detail for a face that has a different shape, or multiple views, because all generated local synthesis faces have the same fixed patch (or template) centralized location, regardless of their shape characteristics. The challenge becomes even greater with these shapes when the face is under extreme poses. A texture can be defined as a function of the spatial variation of brightness intensity of pixels in an image. Each texture level represents a variable, with variations such as smoothness, coarseness, regularity, *etc.*, of each surface oriented in different directions. Our work focuses on two important phenomena: rotation and noise "noise is a term used to describe image information that varies randomly in brightness or color". As a result, if the methods used to eliminate these common phenomena are unreliable, the results will be less accurate; therefore, in practice, the methods used to create the images should be as robust and stable as possible. In addition, the images may differ in position, viewpoint, and light intensity, all of which can influence the final results, challenging texture detail capturing. In order to overcome these challenges, we must adapt a method that can capture and restore the missing texture information. The key to solving this problem is a landmark localization method based on regression. Our work utilized Face Alignment Network (FAN). A FAN framework is based on the HourGlass (HG) architecture, which integrates four Hourglass models to model human pose through hierarchical, parallel, and multi-scale integration to improve texture maps by reconstructing self-occluded parts of faces. The landmark detection algorithm captures and restores the texture details of the synthesised face image by repositioning the

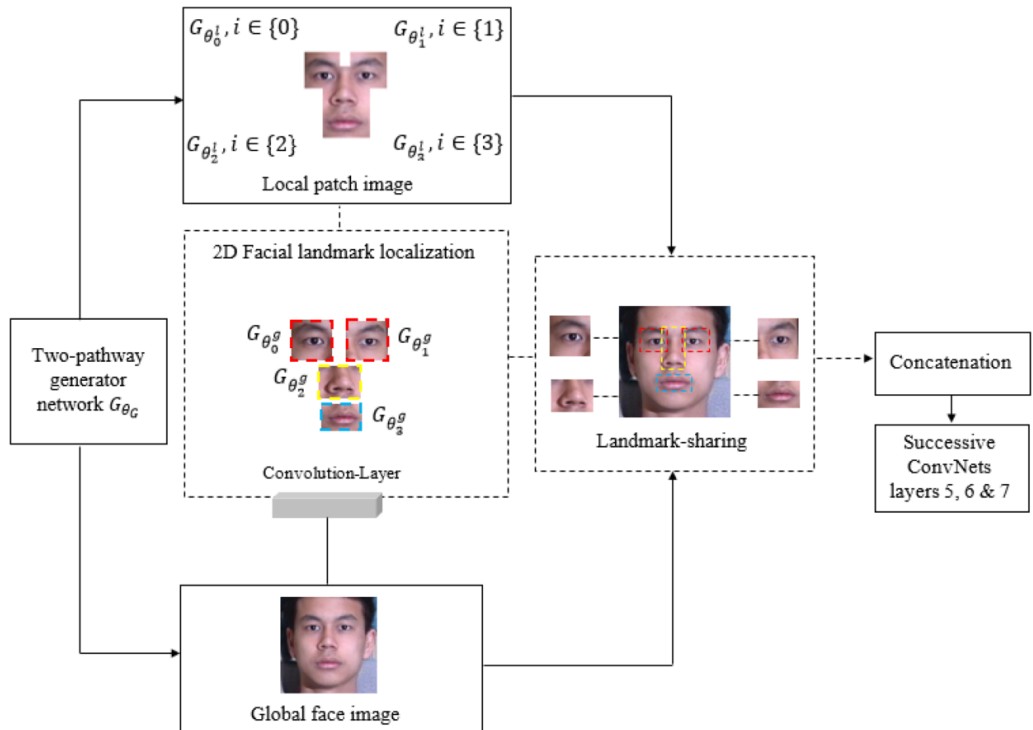

**Figure 2 The structure of the 2D facial landmark localization method.** The dataset was downloaded from the official TP-GAN GitHub page: https://github.com/HRLTY/TP-GAN.

appearance spot of the mismatched or drifted patch "template landmark location". Figure 3 illustrates some examples. Our method allows us to treat faces that have a variety of shape characteristics. In this way, the spatial variations, smoothness, and coarseness that arise due to mismatched or drifted pixels between local and global synthesized faces are eliminated. Typical landmark templates are approximately the same size as a local patch network, but each region has its own structure, texture, and filter. Next, we combined the local synthesis image with the global synthesis image (or two textures) for data augmentation. Every patch of our FAN has its own augmented channels, and each patch has its own RGB along with a depth map (D) input for each 2D local synthesis image. In this way, the texture details help us to build a more robust model around the face patch region and enhance generalization. Even though landmark feature extraction may result in some incongruous or over-smoothing due to noise, it still remains an important method for incorporating pose information during learning.

The landmark detection algorithm for our synthesis face image was built using 68 points. We then reconstructed those synthesis image into four uniform patches (or templates), $Leye = G_{\theta_0^g}, i \in 0$, $Reye = G_{\theta_1^g}, i \in 1$, $Nose = G_{\theta_2^g}, i \in 2$ and $Mouth = G_{\theta_3^g}, i \in 3$, and each patch is comprised of convolutional components. Each patch region has its own filter, which contains different texture details, regain size and structure information. Individual filters provide more details about specific areas in an image, such as pixels or small areas

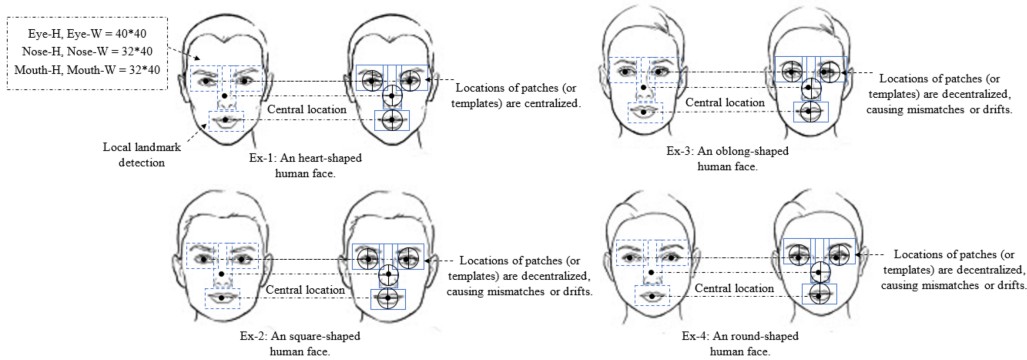

**Figure 3 The detection of different facial shapes characteristics.** In example 1, the appearance of the face is very accurate because all of the generated local synthesis patches (or templates) are centralized. In examples 2, 3, and 4 the appearance of the face is inaccurate since all of the generated local synthesis patches (or templates) are decentralized, resulting in mismatches or drifts. Faces in examples 2, 3, and 4 can be treated using our method regardless of their shape.

**Table 1 An overview of the workflow of the 2D facial landmarks localization including all its operational steps.** These steps are the key to our method that results in a successful implementation.

| Steps | Process |
|---|---|
| 2D landmark detection | 1-Face detection: the goal of this step is to identify faces that are generated by local and global pathways. |
| | 2-Facial landmarks such as the eye centers, tip of the nose, and mouth are located. |
| | 3-The feature extractor encodes identity information into a high-dimension descriptor. |
| Convolutional neural network | 4-The purpose of this layer is to coordinate and extract intermediate features. |
| Data augmentation | 5-This technique is used to enhance the synthesis image textures detail by adding slightly modified copies of already existing data or by creating new synthetic data based on existing data. |
| 5, 6 and 7 layers | 6-5, 6 and 7 act as a visual feature map for specific inputs of fontal-face images in order to retain more visual information by subsampling layers' structure. |

with a high contrast or that are different in color or intensity from the surrounding pixels or areas. Then, one layer of convolutional neural networks is used to coordinate and extract intermediate features. For our method to work more effectively, we remove layers 5, 6, and 7 from ConvNet $\{G_{\theta_D^g}\}$ and replace them after the concatenation stage. Those layers' act as a visualization feature map for a specific input of a fontal-face image in order to increase the amount of visual information kept by subsampling layers' structure. Then, we merged the features tensors of the local and global pathways into one tensor to produce the final synthesis face. Table 1 shows the workflow of all operational steps. The landmark method provides useful information for large-pose regions, *e.g.*, 90°, which helps our model to produce more realistic images.

## Global pathway encoder-decoder structure

In this section we describe our encoder–decoder formulation. Inspired by the work of *Jimei et al. (2016)*, for the DAE, our aim is to train the encoder–decoder with multiple noise levels in order to learn more about the missing texture face details of the input face image and preserve the identity of the frontal-view image $I^F$ from the profile image $I^P$.

The encoder–decoder mechanism has to discover and capture information between the dimensions of the input in order to infer missing pieces and recover the denoised version. In a subsequent paragraph, we will discuss encoder–decoders in more technical details. The idea starts with assuming that the input data points (image pixels) lies on a manifold in $\mathbb{R}^N$. Adding noise to the data image pixel results in diffusing away from the manifold. When we apply DAE to the diffused data image pixels, it tries to pull the data point back onto the manifold. Therefore, DAE learns a vector field pointing from locations with no data point back to the data manifold, implying that it implicitly learns the statistical structure of the data. However, a sparse coding model has been shown to be a good model for image denoising. We assume that group sparse coding, which generalizes standard sparse coding, is effective for image denoising as well, and we will view from a DAE perspective. The encoding function of sparse coding occurs in the inference process, where the network infers the latent variable $s$ from noisy input $\hat{x}$. Each individual symbol is defined here. Let $f_e$ be the RGB components of the input face image, $\Phi$ is the method that splits the input image into its (RGB) components, $\wedge$ is a set of weights and bias for the DAE, and $a$ is the activation function. In our case, the iterative shrinkage-thresholding algorithm (ISTA) is used to perform inference and is formulated as follows:

$$s = f_e(\tilde{x}; \{\Phi, \wedge, a\}) = \text{ISTA}(\tilde{x}; \Phi, \wedge, a). \tag{1}$$

The decoding function is the network's reconstruction of the input from the latent variable.

$$\hat{x} = f_d(s; \Phi) = \Phi_s \tag{2}$$

where $f_d$ is a denoising function.

The DAE method can be used to learn $\wedge$ through the following. For each input data point as shown in Fig. 4, we construct a noisy input by adding Gaussian white noise (GWN) to the original input profile images, as given in: $\tilde{x} = x + v$, where $v \sim N(O, \sigma^2 I)$. Here, $I$ is an $N \times N$ identity matrix, where $N$ is the size of input data (batch output of the generator), and $\sigma^2$ is the noise variance which is the same in all directions. We use group sparse coding to denoise $\tilde{x}$, as described in Eqs. (1) and (2). The DAE is formulated as follows:

$$E_{\text{DAE}} = \|x - \hat{x}\|_2^2. \tag{3}$$

We define DAE's reconstruction error as a square error between $x$ and $\hat{x}$.

Generally, the cost function can be another form of differentiable error measure, shown as follows:

$$\Delta \wedge \propto -\partial E_{\text{DAE}}/\partial \wedge \tag{4}$$

where $\partial E_{\text{DAE}}/\partial \wedge \longrightarrow$ differentiate DAE's reconstruction error with respect to weights, and $\Delta \wedge$ is the change in weights, $\wedge$ can then be learned by doing gradient descent on the DAE's cost function.

Essentially, the $G_{\theta_E^g}$ encodes input data $x \in \mathbb{R}^N$ into a hidden representation: $h = f_e(x; \theta) \in \mathbb{R}^M$. $f_e$ is the encoder and $\theta$ is learning parameters of the DAE function.

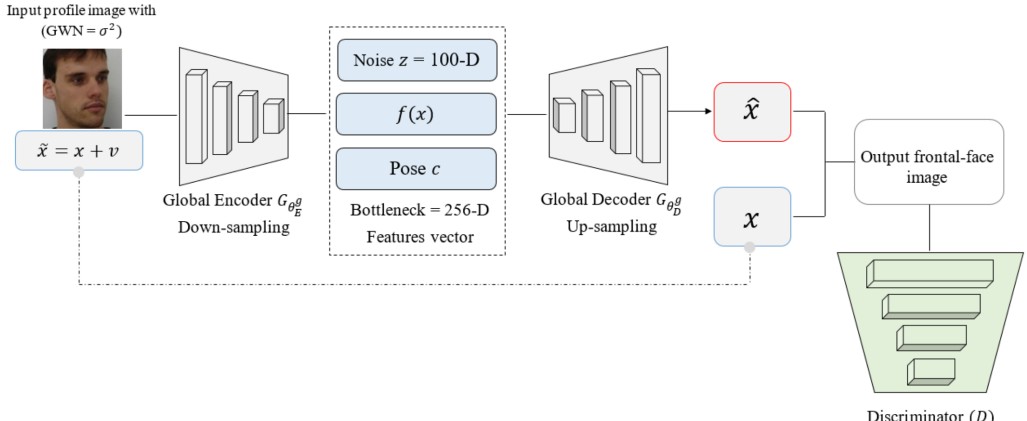

**Figure 4 The global pathway structure. The structure is based on encoder down-sampling and decoder up-sampling. The bottleneck with 256-D features vector remains the same.** The dataset was downloaded from the official FEI website: https://fei.edu.br/~cet/facedatabase.html.

Then, it decodes $G_{\theta_D^g}$ the hidden representation into a reconstruction of the input data: $\hat{x} = f_d(h;\theta)$. The objective for learning the parameter $\theta$ of an $\{G_{\theta_E^g}, G_{\theta_D^g}\}$ is to minimize the reconstruction error between $\hat{x}$ and $x$. Usually, there is some constraint on $\{G_{\theta_E^g}, G_{\theta_D^g}\}$ to prevent it from learning an identity transformation. For example, the dimension of $h$ is much smaller than the input data's dimension $i.e. M \ll N$, then $\{G_{\theta_E^g}, G_{\theta_D^g}\}$ will function similarly to principle component analysis (PCA). The hidden representation $\|h\|_1$ is small, then $G_{\theta_E^g}$ functions similarly to sparse coding. The DAE tries to remove noise from input data. Let $\tilde{x} = x + v$ be a noisy input by adding $v$ to an original input $x$. DAE takes $\tilde{x}$ as an input, then outputs a denoising signal $\hat{x} = f_d(f^e(\tilde{x};\theta);\theta)$. The objective function of a such technique is to minimize the error between $x$ and $\hat{x}$ by adjusting $\theta$, $i.e.,$ it tries to reconstruct the actual content well while not reconstructing noise. DAE can also be viewed as a generative model.

## Adversarial networks

Following (*Ian et al., 2014*) work, adversaries network consists of two components $(G)$ and $(D)$. "The loss function reflects the difference in distribution between the generated and original data". We will first review some technical aspects of the training process, and then the adversaries' network. In frontal view synthesis, the aim is to generate a photorealistic and identity-preserving frontal view image $(I^F)$ from a face image under a different pose, $i.e.,$ a profile image $(I^P)$. During the training phase of such networks, pairs of corresponding $\{I^F, I^P\}$ from multiple identities $y$ are required. Input $I^P$ and output $I^F$ are both based on a pixel space of size $W \times H \times C$ with a color channel $C$. We aim to learn a synthesis function that can output a frontal view when given a profile image. This section will be omitted since it was already explained in TP-GAN architecture. Optimizing the network parameters $(G_{\theta_G})$ starts with minimizing the specifically designed synthesis loss $(L_{\text{syn}})$ and the aforementioned $L_{\text{cross-entropy}}$. For a training set with $N$ training pairs of $\{I_n^F, I_n^P\}$, the

optimization problem is expressed as follows:

$$\hat{\theta}_G = \frac{1}{N} \underset{\theta_G}{\text{argmin}} \sum_{n=1}^{N} \left\{ L_{\text{syn}}\left(G_{\theta_G}(I_n^P), I_n^F\right) + \alpha L_{\text{cross-entropy}}\left(G_{\theta_E^g}\left(I_n^p\right), y_n\right) \right\} \tag{5}$$

where $\alpha$ is a weighting parameter and $L_{\text{syn}}$ is a weighted sum of individual losses that together constrain the image to reside within the desired manifold. Each individual loss function will be explained in the comprehensive loss functions section.

In order to generate the best images, we need a very good generator and discriminator. The reason for this is that if our generator is not good enough, we won't be able to fool the discriminator, resulting in no convergence. A bad discriminator will also classify images that make no sense as real, which means our model never trains, and we never produce the desired output. The image can be generated by sampling values from a Gaussian distribution and feeding them into the generator network. Based on a game-theoretical approach, our objective function is a minimax function.

$$\underset{\theta_G}{\min}\,\underset{\theta_D}{\max}\left[\mathbb{E}_{I^F \sim P(I^F)}\log D_{\theta_D}\left(I^F\right) + \mathbb{E}_{I^P \sim P(I^P)}\log\left(1 - D_{\theta_D}\left(G_{\theta_G}\left(I^P\right)\right)\right)\right] \tag{6}$$

where $I^F$ presents as real frontal face images

$D_{\theta_D}$ presents as the discriminator

$G_{\theta_G}$ presents as the generator

$G_{\theta_G}\left(I^P\right)$ presents as synthesized frontal face images.

Using the discriminator to maximize the objective function allows us to perform gradient descent on it. The generator tries to minimize its objective function, so we can use gradient descent to compute it. In order to train the network, gradient ascent and descent must be alternated.

$$\underset{\theta_D}{\max}\left[\mathbb{E}_{I^F \sim P(I^F)}\log D_{\theta_D}\left(I^F\right) + \mathbb{E}_{I^P \sim P(I^P)}\log\left(1 - D_{\theta_D}\left(G_{\theta_G}\left(I^P\right)\right)\right)\right]. \tag{7}$$

Gradient ascent on $D$.

$$\underset{\theta_G}{\min}\,\mathbb{E}_{I^P \sim P(I^P)}\log\left(1 - D_{\theta_D}\left(G_{\theta_G}\left(I^P\right)\right)\right). \tag{8}$$

Gradient descent on $G$.

Minimax problem allows discriminate to maximize adversarial networks, so that we can perform gradient ascent on these networks; whereas generator tries to minimize adversarial networks, so that we can perform gradient descent on these networks. In practice, Eq. (6) might not provide enough gradients for $G$ to learn well. During the early stages of learning, when $G$ is poor, $D$ can reject samples with a high degree of confidence, since they are clearly different from the training data. In this case, $\log(1 - D_{\theta_D}(G_{\theta_G}\left(I^P\right)))$ saturates. As an alternative to training $G$ to minimize $\log(1 - D_{\theta_D}(G_{\theta_G}\left(I^P\right)))$, we can train $G$ to maximize $\log D_{\theta_D}\left(I^F\right)$. As a result of this objective function, the discriminator and generator have much stronger gradients at the start of the learning process.

When a single discriminating network is trained, the refiner network will tend to overemphasize certain features in order to fool the current discriminator network, resulting in drift and producing artifacts. We can conclude that any local patch sampled from the

refined image should have similar statistics to the real image patches. We can therefore define a discriminator network that classifies each local image patch separately rather than defining a global discriminator network. The division limits both the receptive field and the capacity of the discriminator network, as well as providing a large number of samples per image for learning the discriminator network. The discriminator in our implementation is a fully convolutional network that outputs a probabilistic $N \times N$ ("$N = 2$" represented as a synthesized face image) map instead of one scalar value to distinguish between a ground truth frontal view ($GT$) and a synthesized frontal view ($SF$). Our discriminator loss is defined through the discrepancy between the model distribution and the data distribution, using an adversarial loss ($L_{\text{adv}}$). By assigning each probability value to a particular region, the $D_{\theta_D}$ can now concentrate on a single semantic region rather than the whole face.

## Comprehensive loss functions

In addition to the existing TP-GAN synthesis loss functions, which are a weighted sum of four individual loss functions ($L_{px}$, "$L_{\text{sym}}, L_{ip}, L_{tv}$ and $L_{\text{local}}$"), a classification loss $L_{\text{classify}}$ is further added to our method in order to get better results. Each individual loss function is presented below, and the used symbols are defined here. Let $I$ be an output image.

Let $W$ and $H$ be the width and height of $I$, $(x,y)$ be a pixel coordinate of a 2D image, $I^{\text{pred}}$ be the predicted (*i.e.*, synthesized) frontal-face image of $I$, $I^{gt}$ be the representative frontal-face image of the ground-truth category of $I$. $I^{gt}_{\text{local}}$ be the composition image of the four local facial patches of $I^{gt}$, and $I^{gt}_{\text{local}}$ be the composition image of the four local facial patches of $I$.

## Pixel-wise loss ($L_{px}$)

$$L_{px} = \frac{1}{W \times H} \sum_{x=1}^{W} \sum_{y=1}^{H} |I^{gt}(x,y) - I^{\text{pred}}(x,y)|. \tag{9}$$

Although pixel wise loss may bring some over-smooth effects to the refined results, it is still an essential part for both accelerated optimization and superior performance.

## Symmetry loss ($L_{\text{sym}}$)

$$L_{\text{sym}} = \frac{1}{W/2 \times H} \sum_{x=1}^{W/2} \sum_{y=1}^{H} |I^{\text{pred}}(x,y) - I^{\text{pred}}(W - (x-1),y)|. \tag{10}$$

$L_{\text{sym}}$ is used to calculate the symmetry of the synthesized face image because a face image is generally considered to be a symmetrical pattern.

## Identity preserving loss ($L_{ip}$)

$$L_{ip} = \sum_{i=1}^{2} \frac{1}{W_l \times H_l} \sum_{x=1}^{W_l} \sum_{y=1}^{H_l} \left| F_l^{gt}(x,y) - F_l^{\text{pred}}(x,y) \right|. \tag{11}$$

where $F_l^{\text{gt}}$ and $F_l^{\text{pred}}$ respectively denote the feature map of the last two layers of the light-CNN net (*Xiang et al., 2015*) of $I^{\text{gt}}$ and $I^{\text{pred}}$, $W_l$ and $H_l$ are the width and height of the feature map of the last $l$-th layer ($l = 1, 2$) of the light-CNN net. It is expected that a good synthesized frontal-face image will have similar characteristics to its corresponding real frontal-face image. We employ a fully connected layer of the pre-trained light-CNN net for the feature extraction of the pre-trained recognition network. The pre-trained model will leverage the loss to enforce identity-preserving frontal view synthesis.

## Adversarial loss ($L_{\text{adv}}$)

$$L_{\text{adv}} = \frac{1}{N}\sum_{n=1}^{N} -\log D_{\theta_D}\left(G_{\theta_G}\left(I_n^P\right)\right). \tag{12}$$

$L_{\text{adv}}$ is used to make the real frontal-face image $I^F$ and a synthesized frontal face images $G_{\theta_G}(I_n^P)$ indistinguishable, so that the synthesized frontal face image achieves a visually pleasing effect.

## Total variation loss ($L_{tv}$)

Generally, the face images synthesized by two pathways generative adversarial networks have unfavorable visual artifacts, which deteriorates the visualization and recognition performance. Imposing $L_{tv}$ on the final synthesized face images can help to alleviate this issue. The $L_{tv}$ loss is calculated as follows:

$$L_{tv} = \sum_{x=1}^{W}\sum_{y=1}^{H}\left|I_{x,y}^{\text{pred}} - I_{x-1,y}^{\text{pred}}\right| + \left|I_{x,y}^{\text{pred}} - I_{x,y-1}^{\text{pred}}\right|. \tag{13}$$

$L_{tv}$ will generate a smooth synthesized face image.

## Classification loss ($L_{\text{classify}}$)

$$L_{\text{classify}} = -\sum_{i} y_i^{\text{true}}\log_2\left(y_i^{\text{pred}}\right) \tag{14}$$

where $i$ is the class index, $y^{\text{true}}$ denotes the tensor of the one-hot true target of $I$, and $y^{\text{pred}}$ is the predicted probability tensor. $L_{\text{classify}}$ is a cross-entropy loss which is used to ensure the synthesized frontal-face image can be classified correctly.

## Local pixel-wise loss ($L_{\text{local}}$)

$$L_{\text{local}} = \frac{1}{W \times H}\sum_{x=1}^{W}\sum_{y=1}^{H}\left|I_{\text{local}}^{gt}(x,y) - I_{\text{local}}^{\text{pred}}(x,y)\right|. \tag{15}$$

This loss is to calculate the total average pixel difference between $I_{\text{local}}^{gt}$ and $I_{\text{local}}^{\text{pred}}$, and it is not addressed in *Rui et al. (2017)* work.

**Table 2** **The structure and parameters setting in the LFMTP-GAN.** Those values contribute to the learning process and help our model to achieve its goals.

| Parameters name | Corresponding value |
|---|---|
| Batch Size ($G_{\theta_G}$) | 4 |
| Batch Size ($D_{\theta_D}$) | 10 |
| Epoch Steps | 4500+ |
| Learning Rate ($lr$) | 0.01 |

## Generator loss functions ($L_{\text{generator}}$)

The generator loss function of the proposed method is a weighted sum of all the losses defined above:

$$L_{\text{generator}} = L_{px} + \lambda_1 L_{\text{sym}} + \lambda_2 L_{ip} + \lambda_3 L_{\text{adv}} + \lambda_4 L_{tv} + \lambda_5 L_{\text{classyify}} + \lambda_6 L_{\text{local}} \qquad (16)$$

where $\lambda_i (i = 1 \sim 6)$ are weights that coordinate the different losses, and they are set to be $\lambda_1 = 0.1, \lambda_2 = 0.001, \lambda_3 = 0.005, \lambda_4 = 0.0001, \lambda_5 = 0.1,$ and $\lambda_6 = 0.3$ in our experiments. The $L_{\text{generator}}$ is used to guide an identity-preserving inference of frontal view synthesis. While the $L_{\text{adv}}$ is used to push the generative network forward so the synthesized frontal face image achieves a pleasing appearance.

## Optimizing the training process

In order to optimize the training process, we propose some modifications to the TP-GAN parameters as shown in Table 2 in order to improve the performance in learning the frontal-face data distribution. In our experiment, we consider few parameters: learning rate, batch size, number of epochs, and loss functions. We chose those parameters based on our experience, knowledge and observations. The adopted learning rate improve the module loss accuracy for both $G$ and $D$. The batch size is the number of examples from the training dataset used in the estimation of the error gradient. This parameter determines how the learning algorithm will behave. We have found that using a larger batch size has adversely affected our method performance. As a result, during initial training the discriminator may be overwhelmed by too many examples. This will lead to poor training performance. The number of training epochs is a key advantage of machine learning. As the number of epochs increases, the performance will be improved and the outcomes will be astounding. However, the disadvantage is that it takes a long time to train a large number of epochs. These parameters are essential to improve LFMTP-GAN's representation learning, gaining high-precision performance and reducing visual artifacts when synthesising frontal-face images.

## EXPERIMENTS

We conducted extensive experiments to verify the effectiveness of our method by comparing it with the TP-GAN. The evaluation protocol includes frontal face image resolution and accuracy preserving face identity.

## Experimental settings

Both LFMTP-GAN and TP-GAN models are tested and trained on the Multi-PIE and FEI datasets. Multi-PIE (*Ralph et al., 2010*) is a large face dataset with 75000+ images for 337 identities across a variety of different poses, illuminations and expression conditions captured in a constrained environment. Multi-PIE has 15 poses ranging from ±90°, and 20 illumination levels for each subject. All 20 illuminations were taken within a few seconds: two without any flash illumination, followed by an 18 image with each flash firing independently. Figure 5 shows an example of our model results. "The intensity of light is determined by the brightness of the flash and the background. For example, a bright or dark flash, shadow reflection, or a white or blue background will affect intensity. This depends on the recording equipment and the positioning". To minimize the number of saturated pixels in flash illuminated images, all cameras have been set to have a pixel value of 128 for the brightest pixel in an image without flash illumination. In the same way, the diffusers in front of each flash were added. The color balance was also manually adjusted so that the images looked similar. FEI (*FEI, 2005–2006*) is a Brazilian unlabeled face dataset with 2800+ images for 200 identities across a variety of different poses captured in a constrained environment. The face images were taken between "June 2005 and March 2006" at the artificial intelligence laboratory at são bernardo do campo, são paulo, brazil. The FEI images were taken against a white homogenous background in an upright frontal position with a ±90° range of profile poses; and different illuminations, distinct appearances and hairstyles were included for each subject. Our method shares the same implementation concept as TP-GAN but totally has different parameters settings. The training lasts for 10-to-18 days in each system for each dataset. The training model and source code are available at: https://github.com/MahmoodHB/LFMTP-GAN.

## Visual quality

In this subsection, we compare LFMTP-GAN with TP-GAN. Figures 6 and 7 shows the comparison images, where the first column is the profile-face images under different face poses, the second column is the synthesized frontal-face images by TP-GAN, the third column is the synthesized frontal-face images by LFMTP-GAN, and the last column is one randomly selected frontal-face image of the category corresponding to the profile-face image. The yaw angle of the input face image are chosen circularly from (15°, 30°, 45°, 60°, 75° and 90°). Obviously, the resolution of the LFMTP-GAN images looks better than TP-GAN. This reveals that the use of a proper 2D landmark localization algorithm could significantly improve the image quality of the synthesized images, provide rich textural details, and contain fewer blur effects. Therefore, the visualization results shown in Figs. 6 and 7 demonstrate the effectiveness of our method across a variety of poses and datasets. Similarly, Fig. 8 shows the results of these datasets in close-up image with 90° facial pose.

Previous frontal view synthesis methods are usually based on a posture range of ±60°. It is generally believed that if the posture is greater than 60°, it is difficult to reconstruct the image of the front view. Nonetheless, we will show that with enough training data and a properly designed loss function, this is achievable. In Figs. 9 and 10, we show that LFMTP-GAN can recover identity-preserving frontal faces from any pose, as well as

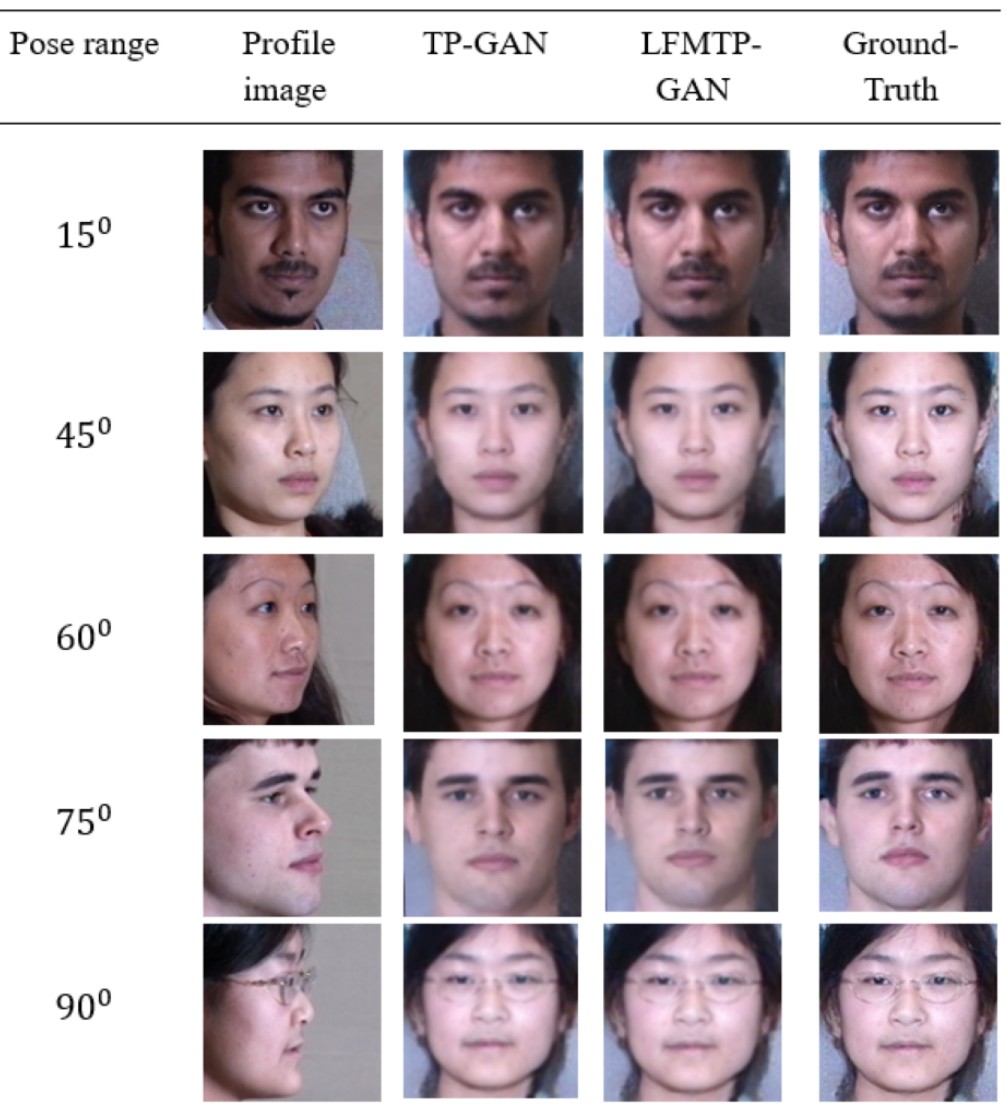

**Figure 5 Examples of different levels of illumination.** The level of illumination, including brightness, exposure, contrast, and shadows, *etc.* Some quality effects can also be observed, such as sharpness, smoothness, blurriness, *etc.* The dataset was downloaded from the official TP-GAN GitHub page: https://github.com/HRLTY/TP-GAN.

comparing with state-of-the-art face frontalization methods, it performs better. In addition, our geometry estimation method does not require 3D geometry knowledge because it is driven by data alone.

## Identity preserving

To quantitatively demonstrate the identity preserving ability of the proposed method, we evaluate the classification accuracy of synthesized frontal-face images on both Multi-PIE and FEI databases, and show their classification accuracy (%) in Table 3 across views and illuminations. The experiments were conducted by first employing light-CNN to

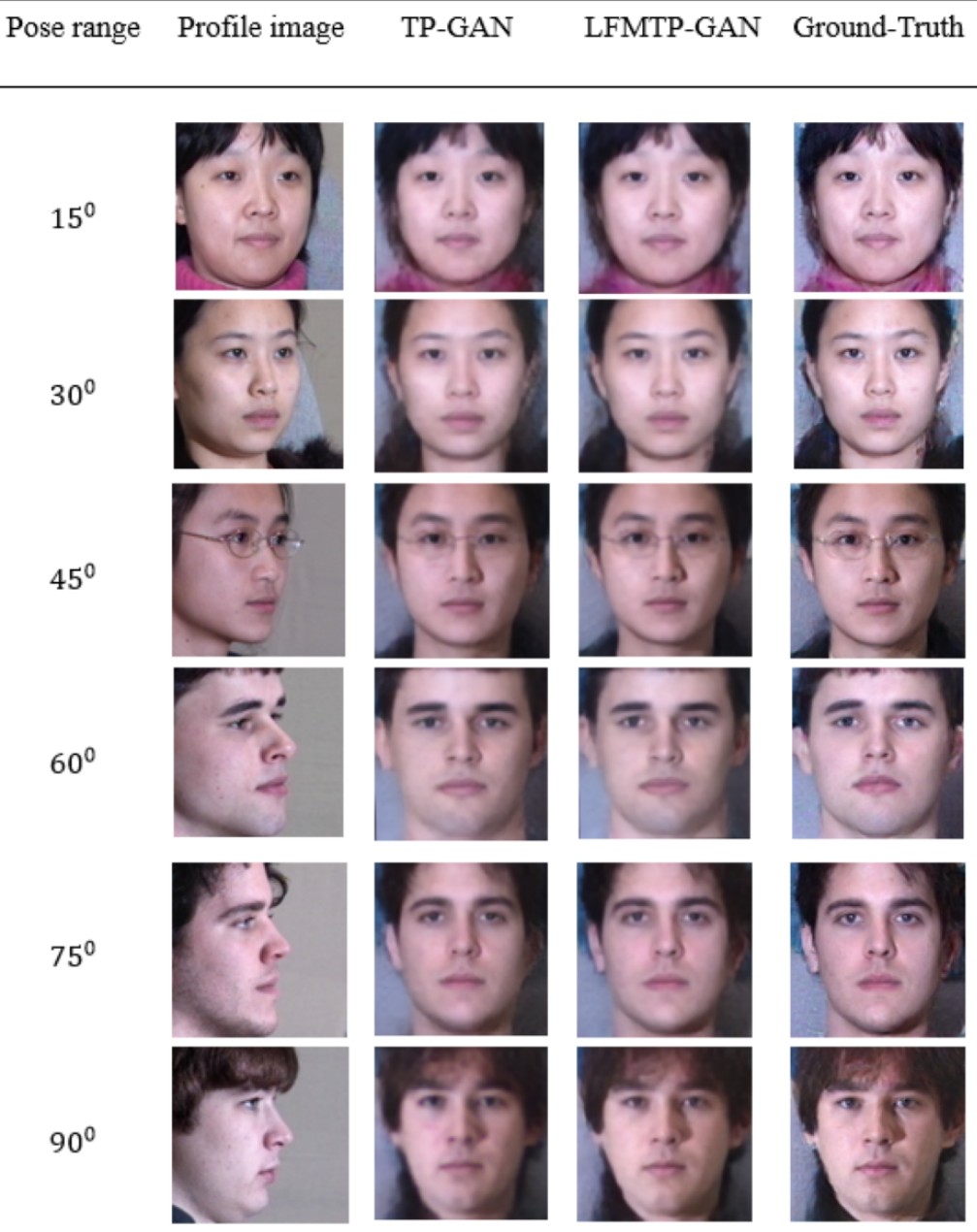

**Figure 6** Comparison of TP-GAN and LFMTP-GAN by synthesizing frontal-face images under different poses and illuminations on Multi-PIE face database. It can be seen that LFMTP-GAN generates better visual quality images. The dataset was downloaded from the official TP-GAN GitHub page: https://github.com/HRLTY/TP-GAN.

extract deep features and then using the cosine-distance metric to compute the similarity of these features. The light-CNN model was trained on MS-Celeb-1M (*Microsoft Celeb, 2016*) which is a large-scale face dataset, and fine-tuned on the images from Multi-PIE and FEI. Therefore, the light-CNN results on the profile images $I^P$ serves as our baseline. Our

**Figure 7** Comparison of TP-GAN and LFMTP-GAN by synthesizing frontal-face images under different poses and illuminations on FEI face database. It can be seen that LFMTP-GAN generates better visual quality images with less blurry visualization effect. The dataset was downloaded from the official FEI website: https://fei.edu.br/~cet/facedatabase.html.

method produces better results than TP-GAN as the pose angle is increased. Our approach has shown improvements in frontal-frontal face recognition. Moreover, Table 4 illustrates that our method is superior to many existing state-of-the-art approaches.

Many deep learning methods have been proposed for frontal view synthesis, but none of them have been proved to be sufficient for recognition tasks. *Chao et al. (2015)* and *Yibo et al. (2018)* relied on direct methods such as CNN for face recognition, which will

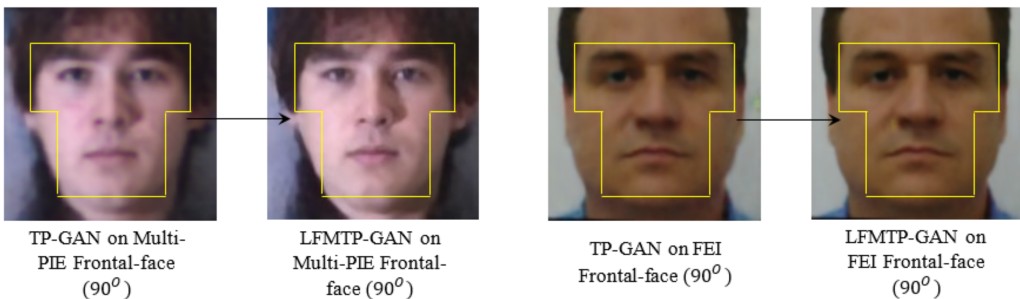

TP-GAN on Multi-PIE Frontal-face (90°) → LFMTP-GAN on Multi-PIE Frontal-face (90°)  TP-GAN on FEI Frontal-face (90°) → LFMTP-GAN on FEI Frontal-face (90°)

**Figure 8** **A comparison of the resolution of facial images taken under 90° face pose, and illuminations condition. The facial visualization area is inside the yellow map.** The dataset was downloaded from the official websites: https://github.com/HRLTY/TP-GAN and https://fei.edu.br/~cet/facedatabase.html.

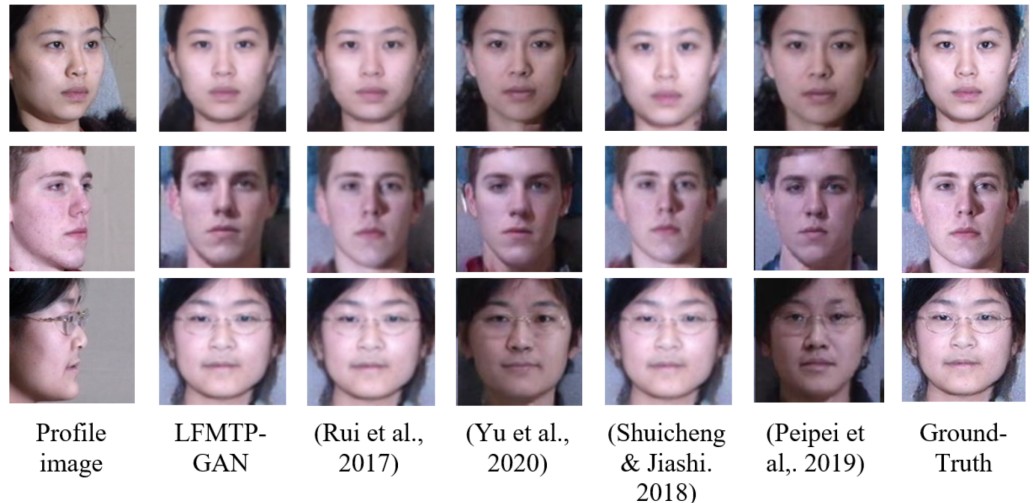

Profile image | LFMTP-GAN | (Rui et al., 2017) | (Yu et al., 2020) | (Shuicheng & Jiashi. 2018) | (Peipei et al,. 2019) | Ground-Truth

**Figure 9** **A comparison of LFMTP-GAN algorithms with different techniques for face frontalization on Multi-PIE dataset.** The dataset was downloaded from the official TP-GAN GitHub page: https://github.com/HRLTY/TP-GAN. The other image used for face recognition comparison was downloaded from: https://github.com/YuYin1/DA-GAN.

definitely reduce rather than improve performance. It is therefore important to verify whether our synthesis results can improve recognition performance (whether "recognition *via* generation" works) or not. The next section presents the loss curves for the two models.

## Model loss curve performance

This section provides a comparison with TP-GAN. We analyze the effects of our model on three tradeoff parameters named generator loss, pixel-wise loss, and identity preserving accuracy. 80% of the face image subjects from the Multi PIE and FEI datasets were used for training and evolution purposes. 90% of the image subjects were used for training, while 10% were used for testing. The recognition accuracy and corresponding loss curves are shown in Fig. 10. We can clearly see from the curves that, the proposed method

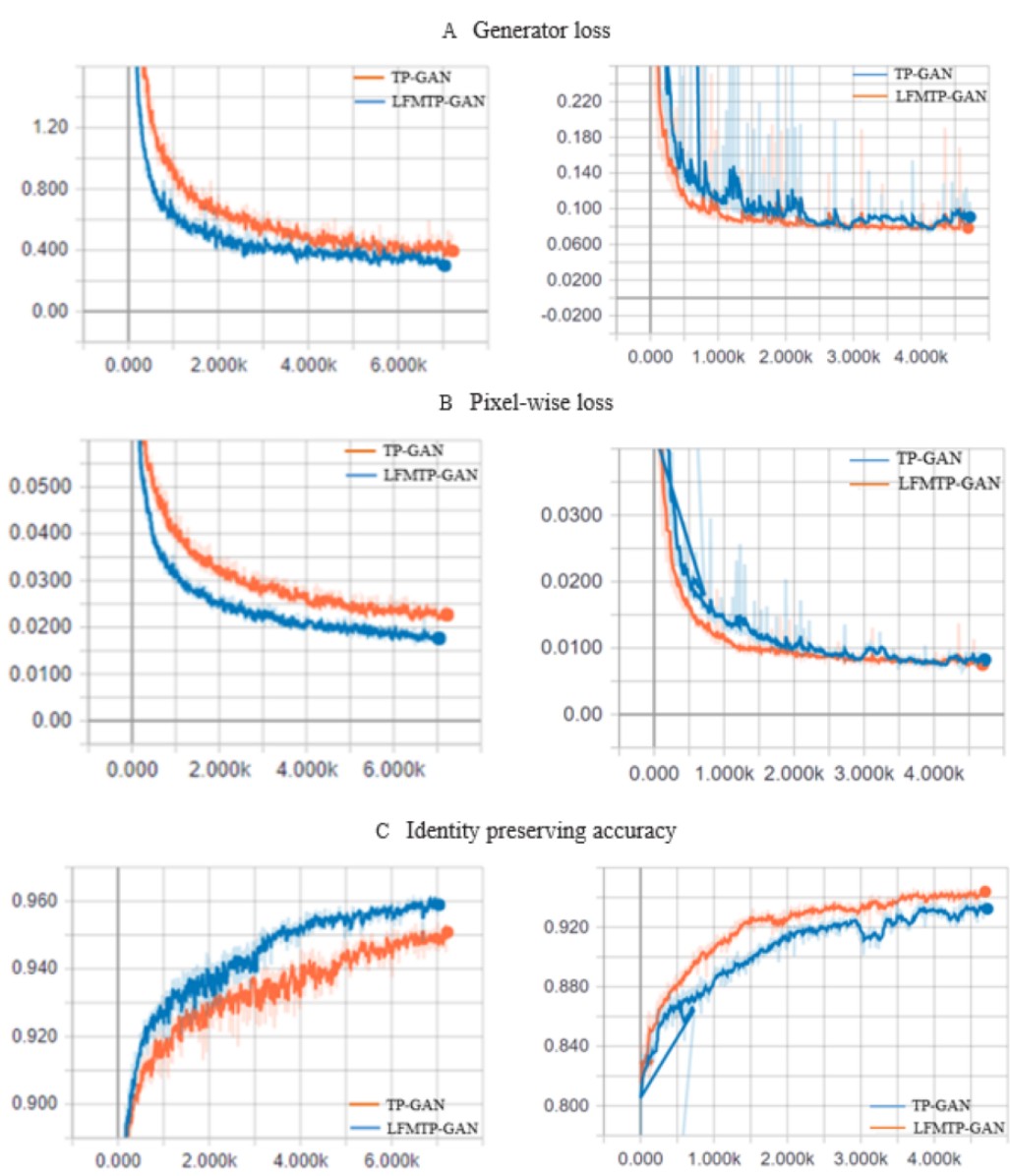

**Figure 10** **The TP-GAN and LFMTP-GAN loss curve plots.** (A) The generator loss curve, and (B) pixel-wise loss curve. The horizontal axis indicates the number of epochs, which is the number of times that entire training data has been trained. The vertical axis indicates how well the model performed after each epoch; the lower the loss, the better a model. (C) The identity preserving accuracy curve, which is a quality metric that measures how accurate it is to preserve a user's identity; the higher the accuracy, the better the model.

improves the TP-GAN model and provides a much better performance on both datasets. In particular, the number of epochs exceeds 4500, the loss performance decreased sharply in LFMTP-GAN model, while the loss performance decreased slightly in the TP-GAN model. Our optimization learning curves was calculated according to the metric by which the parameters of the model were optimized, *i.e.,* loss. More importantly, our method still

**Table 3 Cross-validation of face recognition rates on Multi-PIE and FEI datasets using weights that coordinate different loss value parameters and Table 1 settings.** The run time is measured in minutes.

| Method | Train | Valid | Cross-valid ($L_s$) | Time (m) |
|---|---|---|---|---|
| | | Multi-PIE dataset | | |
| TP-GAN | 95.2% | 91.7% | 0.165 | 373 |
| LFMTP-GAN | 96.1% | 92.7% | 0.164 | 290 |
| | | FEI dataset | | |
| TP-GAN | 94.0% | 90.0% | 0.179 | 303 |
| LFMTP-GAN | 95.0% | 91.3% | 0.164 | 315 |

**Table 4 The recognition rate (%) across views and illuminations based on the Multi-PIE dataset under Table 1 settings.** This table compares different methods of facial recognition. Our method is capable of outperforming and achieving a better result.

| Method Angle | LFMTP-GAN | *Rui et al. (2017)* | *Luan, Xi & Xiaoming (2017)* | *Yibo et al. (2018)* | *Chao et al. (2015)* | *Xiang et al. (2015)* | *Junho et al. (2015)* |
|---|---|---|---|---|---|---|---|
| ±90° | 65.23% | 64.03% | – | **66.05%** | 47.26% | 5.51% | – |
| ±75° | **85.30%** | 84.10% | – | 83.05% | 60.66% | 24.18% | – |
| ±60° | **94.13%** | 92.93% | 83.2% | 90.63% | 74.38% | 62.09% | – |
| ±45° | **98.80%** | 98.58% | 86.2% | 97.33% | 89.02% | 92.13% | 71.65% |
| ±30° | **99.88%** | 99.85% | 90.1% | 99.65% | 94.05% | 97.38% | 81.05% |
| ±15° | 99.80% | 99.78% | 94.0% | **99.82%** | 96.97% | 98.68% | 89.45% |
| Average ACC | **90.523%** | 89.878% | 88.375% | 89.421% | 77.056% | 63.328% | 80.716% |

produces visually convincing results (as shown in Figs. 6, 7 and 9) even under extreme face poses, its recognition performance is about 1.2% higher than that of TP-GAN.

## RESULTS AND DISCUSSIONS

The goal of this method is to match the appearance of each query face by marking the partially face surface of the generated image, such as the eyes, nose and mouth. In theory, this would have allowed the TP-GAN method to better preserve facial appearance in the updated, synthesized views. The statement holds true when the face is considered to have similar shape characteristics. Considering that all human faces have unique shape characteristics, this may actually be counterproductive and harmful, rather than improving face recognition. We believe that it is necessary to integrate the four important local areas (eyes, nose and mouth) into their right positions on the whole face image, which is already proved to be correct from our experiment results. A majority of face recognition systems require a complete image to be recognized, however, recovering the entire image is difficult when parts of the face are missing. This makes it difficult to achieve good performance. We demonstrate how our approach can help enhance face recognition by focusing on these areas of the face and outperform other methods in the same context. Landmark detection is widely used in a variety of applications including object detection, texture classification, image retrieval, *etc*. TP-GAN already has landmark detection implemented for detecting the four face patches in its early stages as shown in Fig. 1. Such a method

might be valuable in obtaining further textual details that can help in recovering those facial areas and face shapes that have different characteristics. In our proposal, we offer a relatively simple yet effective method for restoring the texture details of a synthesised face image by repositioning the appearance areas of four landmark patches rather than the entire face. We show a different method for facial recognition (Table 4) for faces with extreme poses. In four major poses we achieve rank-1 recognition rates (75%, 60%, 45%, and 30%). Furthermore, when it comes to global $\{G_{\theta_E^g}, G_{\theta_D^g}\}$, we found that optimizing for the corrupted images resulted in a better convergence rate than optimizing for the clean images in order to achieve balanced learning behavior. Therefore, we neither extensively modify the $\{G_{\theta_E^g}, G_{\theta_D^g}\}$, nor include an external neural network in our modification, because that integration will increase the network complexity and cause training limitations. We obtained better results by introducing noise to the image before feeding it to the $\{G_{\theta_E^g}, G_{\theta_D^g}\}$ during optimization.

## CONCLUSION

In this paper, we propose an LFM model for synthesizing a frontal-face image from a single image to further enhance the frontal-face images quality of the TP-GAN model. To accomplish our goal smoothly, we expand the existing generative global pathway with a well-constructed 2D face landmark localization to cooperate with the local pathway structure in a landmark sharing manner to incorporate empirical face pose into the learning process, and improve the encoder–decoder global pathway structure for better facial image features representation. Compared with TP-GAN, our method can generate frontal images with rich texture details and preserve the identity information. Face landmark localization allows us to restore the missing information of the real face image from the synthetic frontal images, and provide a rich texture detail. The quantitative and qualitative experimental results of the Multi-PIE and FEI datasets show that our proposed method can not only generate high-quality perceptual facial images in extreme poses, but also significantly improves the TP-GAN results. Although LFMTP-GAN method achieves a high-quality image resolution output, there is still room for improvement by choosing different optimization algorithms, such as loss functions, or introducing some different techniques for facial analysis and recognition. Our future research is to apply different error functions or different face analysis and recognition techniques, combined with two pathway structures, to achieve a super-resolution generative model and high-precision performance.

### Funding

The authors received no funding for this work.

### Competing Interests

The authors declare there are no competing interests.

## Author Contributions

- Mahmood H.B. Alhlffee and Yi-An Chen conceived and designed the experiments, performed the experiments, analyzed the data, performed the computation work, prepared figures and/or tables, authored or reviewed drafts of the paper, and approved the final draft.
- Yea-Shuan Huang conceived and designed the experiments, performed the experiments, analyzed the data, prepared figures and/or tables, authored or reviewed drafts of the paper, and approved the final draft.

## Data Availability

The data is available at GitHub: https://github.com/MahmoodHB/LFMTP-GAN.

## Supplemental Information

Supplemental information for this article can be found online at http://dx.doi.org/10.7717/peerj-cs.897#supplemental-information.

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
