# Peer review of "D facial landmark localization method for multi-view face synthesis image using a two-pathway generative adversarial network approach"

_PeerJ Computer Science, doi:10.7717/peerj-cs.897_

## Round 0.1 · original submission · Major Revisions

In view of the criticism of the reviewers, authors are invited to resubmit a revised version of the manuscript for further consideration in PeerJ Computer Science. In addition to reviewers comments, I also have the following comments for the authors.
- Literature is not well presented in the manuscript and recent studies on 2021 are lacking.
- The proposed architecture is still missing some technical details. For instance, the architectures of generator and discriminator. Same for the training process.
- The baseline performance of the proposed architecture should be explored.
- Some abbreviations are not properly used, e.g. "LFMTP-GAN" and "DAE" are defined twice.

Also, Reviewer 2 has requested that you cite specific references. You may add them if you believe they are especially relevant. However, I do not expect you to include these citations, and if you do not include them, this will not influence my decision.

You may find these publications useful:
10.1109/TIP.2020.2976765
https://doi.org/10.1016/j.patcog.2021.107893
https://doi.org/10.1016/j.neucom.2020.04.025
https://doi.org/10.1016/j.cviu.2020.103036

·

Basic reporting

- Clarity in the overall structure of manuscript sentences.
- Literary study can be further extended as there are no studies referred for the year 2021.
- Sufficiently detailed tables and figures which is a plus point.
- Abstract mentions "improved the encoder-decoder global pathway structure"; however, the abstract fails to mention the details of this improvement in terms of accuracy or operational workflow.
- Authors mention throughout the manuscript the need for the global structure; however, the reader can still do not get the idea of its importance and impact on the overall verification process.
- Line 459 mentions 20 illumination levels for the image dataset. Mention the details of these levels.

Experimental design

- Line 5,6,7 mentions there is a drop of 10% of recognition. Can authors present and validate this point through focused experiments and also show the gain of the proposed methods over the same set of images (for frontal-profile to frontal-frontal face verification)

Validity of the findings

- The results reported are promising. It would be great if the authors could add some more experiments to show the effect of 20 different illumination levels as for angles presented in Table 3.

Additional comments

- Related work should be summarised and a small paragraph should be added at the end of the section to discuss the key takeaway of literary study to facilitate the reader.
- Proposed method should be presented in one concrete table with all the operational steps involved (all in one place)


Overall the research is well-defined and has a high potential to fill the identified research gap.

·

Basic reporting

- English language should be improved. There still have some grammatical errors, typos, or ambiguous parts.

- "Introduction" is verbose, it could be revised to make it more concise.

- Quality of figures should be improved significantly. Some figures had a low resolution.

Experimental design

- Source codes should be provided for replicating the methods.

- Statistical tests should be conducted in the comparison to see the significance of the results.

- The authors should describe in more detail on the hyperparameter optimization of the models.

- Deep learning is well-known and has been used in previous studies i.e., PMID: 31920706, PMID: 32613242. Thus, the authors are suggested to refer to more works in this description to attract a broader readership.

- Cross-validation should be conducted instead of train/val split.

Validity of the findings

- The authors should compare the performance results to previously published works on the same problem/dataset.

- It lacks a lot of discussions on the results.

- The model contained a little bit overfitting, how to explain this case?

Additional comments

No comment.

---

## Round 0.2 · Minor Revisions

The authors have greatly improved the manuscript and satisfactorily responded to reviewers' concerns. Generally, the manuscript is ready for acceptance except for very minor comments as given below.

1. There are many loss functions that are perplexing. Please, tidy up this part to avoid any misunderstanding by readers.
2. There is a typo in line 313 "is the a set".
3. Also, in Line 353 "the set of database".

It would be better to double-check the manuscript against grammatical errors before final submission.

·

Basic reporting

No comment.

Experimental design

No comment.

Validity of the findings

No comment.

Additional comments

No comment.

---

## Round 0.3 · accepted · Accept

The authors have revised the manuscript and it is now ready for publication. Congratulations!